# EncryptedLLM: Privacy-Preserving Large Language Model Inference via GPU-Accelerated Fully Homomorphic Encryption

**Leo de Castro** [1]  **Daniel Escudero** [2]  **Adya Agrawal** [2]  **Antigoni Polychroniadou** [2]  **Manuela Veloso** [2]

## Abstract

As large language models (LLMs) become more powerful, the computation required to run these models is increasingly outsourced to a third-party cloud. While this saves clients' computation, it risks leaking the clients' LLM queries to the cloud provider. Fully homomorphic encryption (FHE) presents a natural solution to this problem: simply encrypt the query and evaluate the LLM homomorphically on the cloud machine. The result remains encrypted and can only be learned by the client who holds the secret key. In this work, we propose a GPU-accelerated FHE scheme and leverage it to benchmark an encrypted GPT-2 forward pass. Our approach achieves runtimes that are over $200\times$ faster than the CPU baseline. We also present novel and extensive experimental analysis of approximations of LLM activation functions to maintain accuracy while achieving this performance.

## 1. Introduction

Large language models (LLMs) have proven to be groundbreaking artificial intelligence tools that are set to change the way humans interact with software. By training on massive amounts of data and using an incredibly large amount of trainable parameters, LLMs are able to provide unprecedented inference results. The tasks that LLMs excel at include natural language generation, question-answering, summarization, translation, code generation, among several others. Models like GPT-3[1] can produce coherent and contextually appropriate text on a wide range of topics. However, these models require massive amounts of resources to be trained, and are often not publicly available as this constitutes the provider's intellectual property. This leads to a "inference-as-a-service" scenario, where clients send their queries to external providers who locally run an LLM to return a result to the client. Furthermore, even open source LLMs such as Llama 2[2] are very expensive to run in commodity hardware and still require in most cases delegating inference to a third party provider.

Unfortunately, delegating inference is undesirable in many settings where the client wants to preserve the privacy of their input. Furthermore, as mentioned above, there are multiple contexts in which the model owner also wants to retain privacy of the model itself, for example when the model involves massive monetary resources to be trained, or when it incorporates sensitive data (*e.g.* a bank servicing a credit score model trained on internal data). This is particularly relevant as LLMs become more pervasive and find more use-cases that permeate all areas of society. This tension between privacy and utility heavily limits the applicability of LLMs, rendering them useless in contexts where data cannot be outsourced due to privacy constraints.

Towards resolving this tension, fully homomorphic encryption (FHE) is a promising tool that enables computing on data without revealing it, only outputting the final result (*cf* (Marcolla et al., 2022) for a survey). Using FHE, a client can *encrypt* their query to the server, who can locally apply their model to this encrypted data, making use of the homomorphic properties of the scheme to obtain an encrypted result, which is sent back to the client for decryption. See Fig. 1a for a pictorial representation of this interaction pattern. Advances in the last decade on all fronts including algorithms, software, and hardware, have made FHE practical for several tasks that were not within reach before. However, LLMs are in an entirely different regime: their computation is already very expensive in the clear, up to the point in which specialized software such as high-end GPUs, coupled with several architectural optimizations, are needed in order to provide a reasonable inference latency. Any computation that is ran under FHE becomes *much* slower, which is going to be a major blocker when porting LLMs to FHE. However, the question remains:

---

[1]J.P. Morgan Chase Cybersecurity & Technology Controls, New York, New York, USA [2]J.P. Morgan AI Research & AlgoCRYPT CoE, New York, New York, USA. Correspondence to: Leo de Castro <leo.decastro@jpmchase.com>.

*Proceedings of the 42nd International Conference on Machine Learning*, Vancouver, Canada. PMLR 267, 2025. Copyright 2025 by the author(s).

[1]https://openai.com/index/gpt-3-apps/

[2]https://llama.meta.com/llama2/

*How practical is FHE-based privacy-preserving LLM evaluation?*

**Approximating LLM Activation Functions.** Fully homomorphic encryption can, in principle, evaluate any function over encrypted data. One approach to implementing an LLM in FHE is to simply evaluate the exact Boolean circuit the describes the LLM over the encrypted input. However, the vast majority of operations in an LLM are the additions and multiplications the comprise the linear layers. Converting arithmetic operations to Boolean operations results in a significant overhead. This is particularly true in modern FHE schemes, which naturally support the arithmetic operations of addition and multiplication over encrypted vectors. Despite comprising the majority of the work in plaintext LLM evaluation, the linear operations in encrypted LLM evaluation are quite cheap. Instead, the majority of our runtime comes from evaluating the LLM activation functions. As is the case when representing arithmetic operations with Boolean gates, representing the complete activation functions with arithmetic gates would result in substantial overhead. Instead, we opt for evaluating only *approximations* of the LLM activation functions. Intuitively, this works because modern LLMs perform well even with low precision, which has been demonstrated in the success of quantized LLM models. We provide extensive accuracy benchmarks of our LLM with approximate activation functions in section 4.

To address this question, a good starting point is the CKKS scheme by (Cheon et al., 2017), which enables approximate additions and multiplications over real (in fact, complex) numbers. We provide a general introduction to FHE in section 2.2. The literature in improving the efficiency of this scheme is vast and fruitful (Han & Ki, 2020; Bossuat et al., 2021; Jung et al., 2021), and this has enabled several applications in contexts such as logistic regression (Chen et al., 2018a) and secure password search (Chen et al., 2018b).

Only the recent work of (Zhang et al., 2024) has explored large language model inference via CKKS, reporting an implementation of the transformer architecture in C++, using the SEAL library for FHE (https://github.com/Microsoft/SEAL). Their experiments report minor accuracy degradation due to polynomial approximations needed in FHE, and performance in Intel CPUs seems promising, as it is accelerated via HEXL (Boemer et al., 2021). We discuss this work further in section 1.2. Although promising given the massive overheads involved in both LLMs and FHE, this is still far from practical for real-world usage, even for applications that are not latency sensitive such as text summarization or content generation (in contrast to chatbots or Q/A tasks, which are more demanding in terms of responsiveness).

## 1.1. Our contributions

We approach the problem of improving the efficiency of FHE-based privacy-preserving LLM inference, by providing a *GPU-based* implementation of the transformer architecture using CKKS. Prior work (Jung et al., 2021) has shown GPUs to help in improving the efficiency of CKKS. However, to the best of our knowledge, there is currently no available implementation of such works to deploy and test these ideas. In contrast, there are popular open-source *CPU-based* frameworks that aim at making FHE techniques more accessible by providing high level programming interfaces, and access to multiple FHE schemes, like CKKS. One such framework is *OpenFHE* (Al Badawi et al., 2022), which has gained traction as one of the most comprehensive and widely used FHE implementations available. Unfortunately, OpenFHE is limited to CPUs, and hence its performance in tasks such as LLM inference would be rather poor.

In this work we extend the capabilities of OpenFHE by enabling a GPU-based workflow, which leads to direct efficiency improvements across many FHE applications that build on this framework—not only LLMs. This involves combining several optimizations (de Castro et al., 2021; Kim et al., 2022) to simultaneously achieve high performance and high accuracy in this implementation. We have open-sourced the code of our OpenFHE+GPU extension [3], which we believe will be of independent interest. To our knowledge, this is the fastest open-sourced implementation of CKKS when running on a GPU.

With our GPU-based implementation in place, we set out to benchmark the performance of large language models under FHE. We focus specifically on the GPT-2 architecture by OpenAI, which is fully open-source and shares common features with many of the more powerful industry-grade models. One first obstacle we face is that FHE techniques do not support all operations available to a common CPU/GPU and instead only supports additions and multiplications. As usual in the FHE literature, we use off-the-shelf polynomial approximations to replicate as faithfully as possible the transformer architecture, while adapting for FHE use. Notably, we incorporate the recent Soft-Max optimization suggestion of (Cho et al., 2024), which substantially improves the runtime of the approximation by removing the expensive computation of the max circuit and replacing this circuit with just a table lookup. This required computing the table of max lookup values, which we based on the extensive tests of the approximate model. Our approximations are discussed in Section 3. Note that these modifications have the potential of negatively affecting the accuracy of the model, which is far from

---

[3] https://github.com/leodec/openfhe-gpu-public

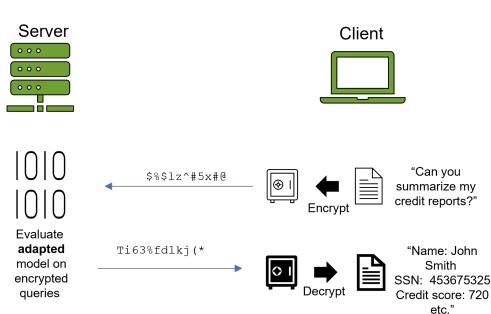

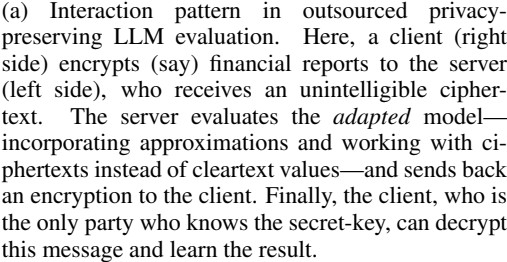

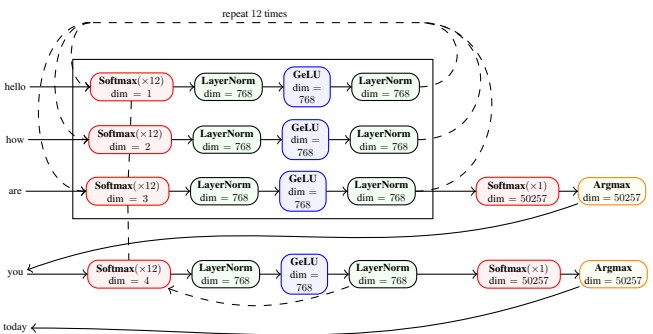

(a) Interaction pattern in outsourced privacy-preserving LLM evaluation. Here, a client (right side) encrypts (say) financial reports to the server (left side), who receives an unintelligible ciphertext. The server evaluates the *adapted* model—incorporating approximations and working with ciphertexts instead of cleartext values—and sends back an encryption to the client. Finally, the client, who is the only party who knows the secret-key, can decrypt this message and learn the result.

(b) Overview of the GPT-2 inference flow. The diagram only shows the blocks that are expensive to run in FHE, ignoring simpler operations such as linear or affine layers. The input is the sentence "hello how are", and the completion is "you". To get the next token, only the path associated to the lastest token "you" needs to be computed, which in this example leads to the token "today". The vertical lines between the leftmost softmax boxes illustrates that each new softmax is somehow dependent on the inputs of previous ones. Every block is labeled with the dimension of the input it takes. For softmax, the number in parentheses represents how many such calls are made.

*Figure 1.* On the left: communication pattern between the two parties. On the right: GPT-2 architecture, which corresponds to the local computation by the server.

ideal. To address this, we modify the GPT-2 implementation from HuggingFace's transformers library (https://github.com/huggingface/transformers) so that it includes these FHE-friendly modifications, and thoroughly benchmark the resulting accuracy using the LM evaluation harness library (https://github.com/EleutherAI/lm-evaluation-harness) on a selection of tasks. This allows us to select optimal parameters for the approximations that strike the right balance between efficient FHE runtimes and model accuracy. Furthermore, for reproducibility we also open source our modified HuggingFace GPT-2 implementation.

Our results given in section 4 show that a GPU-accelerated FHE implementation provides a roughly $200\times$ speedup in the GPT-2 forward pass, reducing the time from several hours to just a few minutes. This brings the forward pass time down to a range where non-real-time applications become more practical, such as document summarization and fine-tuning models on private data.

### 1.2. Related Work

There is a long line of works studying secure inference for protecting the privacy of both a client owning a query, and a server holding a trained model. At a high level, we can divide these techniques into two groups: highly interactive approaches based on MPC, and less communication-demanding but more computationally-heavy paradigms based on FHE.

**FHE-based LLM inference.** FHE-based secure inference has the notable advantage that it preserves the same communication pattern of non-private inference: the client sends the query to the server, who performs certain (presumably heavy) computation and sends back the result. This is applicable to real-world settings where client and servers may not be well connected, and the server is considerably more powerful than the client. In this context, the most relevant work in secure LLM inference with FHE is (Zhang et al., 2024). This work makes use of several polynomial approximations from the literature, some of which we borrow as well (see Section 3). Importantly, their implementation is limited to CPU, which caps their performance substantially. Rather than comparing to this work, we instead compare directly to the out-of-the-box OpenFHE CPU implementations of the FHE functions. This allows us to account for variations in the approximations and the placement of the bootstrapping functions.

The work of (Zimerman et al., 2023) studies HE-friendly approximation of the transformer architecture, but it is not applicable to our case since this require re-training. Primer (Zheng et al., 2023) and THE-X (Chen et al., 2022) also employ FHE for LLM evaluation (Primer in fact mixes FHE and MPC), but these works also make substantial modifications to the underlying model. THE-X even reveals intermediate values of the computation.

**Privacy-preserving ML for other models.** Finally, we mention that there are several other works that have studied FHE-based inference of other machine learning models,

such as convolutional neural networks (*cf.* (Ao & Boddeti, 2024; Juvekar et al., 2018; Gilad-Bachrach et al., 2016; Boemer et al., 2019)). These are not applicable to transformers directly as they do not support all of the operations involved in this architecture, and additionally the scale of the models they consider is much more reduced.

**LLMs and privacy.** The works of (Mireshghallah et al., 2024; Shao et al., 2024) expose critical privacy risk in LLMs, showing they often disclose sensitive information in inappropriate contexts. In particular, the authors of (Mireshghallah et al., 2024) introduce CONFAIDE, a benchmark assessing LLMs' privacy reasoning, revealing that even GPT-4 and ChatGPT fail to uphold contextual privacy 39% and 57% of the time. This highlights the urgent need for stronger privacy-preserving approaches, such as performing LLM computations on encrypted data without decryption and ensuring that only authorized parties can decrypt and access the final outputs.

### 1.3. Setting and Threat Model

We consider a client who holds as input a text sequence, and a server who holds a large language model. The goal is for the client to learn the evaluation of their query on the model without leaking the input to the server, and while protecting the privacy of the model towards the client. See Fig. 1a for a pictorial representation of the task and the communication flow. The server does not learn any information about the client's input, but we provide no correctness guarantees regarding the result the server returns to the client—a corrupt server can return an incorrect answer, or no answer at all. This is consistent with prior works, and it is strictly better than the guarantees provided by MPC-based solutions, which may leak information towards a corrupt server that deviates from the protocol specification.

We assume the client has access to the *tokenizer* of the model so that the client can locally transform their text into a sequence of real-valued vectors, which are then encrypted towards the server. We do not provide any guarantees on the plaintexts underlying the ciphertexts that the client sends. In particular, a corrupt client may send a sequence of vectors that does not correspond to valid token embeddings, and will be able to learn the LLM evaluation on this input. This is in par with previous privacy-preserving ML works based on FHE.

## 2. Preliminaries

In what follows we provide background on large language models and fully homomorphic encryption.

Some general notation we will use throughout the paper is the following. Vectors are denoted by bold letters, like $\boldsymbol{x}$,

and indexing the $i$-th entry is denoted by $\boldsymbol{x}[i]$. Given a positive integer $n$, we let $[n]$ denote the set $\{1, \ldots, n\}$.

### 2.1. Large Language Models

A large language model (LLM) is a type of machine learning (ML) model that is characterized by its ability to predict *language*, with the "large" term emphasizing their comparatively gigantic sizes and computational demands. The work of (Vaswani et al., 2017) introduced the transformer architecture, which is the basis for several LLMs that came right after. Among LLMs, an interesting and relevant family are generative pretrained transformers (GPTs), which are used in natural language processing contexts. This family, developed by OpenAI, has been widely influential and has spawned a series of follow-ups. In this work we focus specifically on the **GPT-2** model, which is trained on Web-Text: 40 GB of text, 8 million documents, from 45 million webpages upvoted on Reddit. We chose this model as (1) it is fully *open source*, (2) it follows the transformer architecture shared by other more powerful LLMs, and (3) this is already challenging in terms of efficiency for current FHE approaches. We note however that our findings carry out to several other LLMs that follow this paradigm, such as the larger models like GPT-3 or GPT-4 or other transformer-based LLMs like Llama and Llama 2. In what follows, we describe the GPT-2 architecture in detail. There are four variants of GPT-2 which vary in size and performance: S, M, L and XL, and we discuss below the points where these differ.

LLMs use deep learning to analyze and generate human-like text. The transformer architecture by (Vaswani et al., 2017) receives as input a piece of text, which is split into numerical representations referred to as *tokens*. Transformers are comprised of an enconder and a decoder section, which are very similar in structure. However, generative LLMs such as GPT are *decoder-only*, and so for the sake of this work we will focus on the decoder component of the transformer architecture; we note that encoders follow a similar structure and our findings apply to encoder-decoder or encoder-only architectures as well.

The model is trained to predict the best next word given a sequence of words. For example, it may receive as an input "Today is a good", and then predict "day" as the next word. The resulting concatenated sentence "today is a good day" can be fed into the model again to obtain as the next word, perhaps, "for". This way a sequence like "today is a good day for running outside" can be generated.

An overview of the GPT-2 architecture, highlighting the blocks that are most relevant for FHE, is given in fig. 1b./

## 2.2. Fully Homomorphic Encryption

A fully homomorphic encryption (FHE) scheme (Rivest et al., 1978), (Gentry, 2009) is an encryption scheme that allows computations to be performed over the data while the data remains encrypted. More formally, an FHE scheme is defined by the following tuple of algorithms.

- $(\mathsf{sk}, \mathsf{pk}, \mathsf{evk}) \leftarrow \mathsf{KeyGen}(1^\lambda)$. This is the key generation algorithm. The input is the security parameter $\lambda$ and the output is three keys. The secret key $\mathsf{sk}$ is used for decryption, the public key $\mathsf{pk}$ is used for encryption, and the evaluation key $\mathsf{evk}$ is used to homomorphically compute over encrypted data.

- $\mathsf{ct} \leftarrow \mathsf{Encrypt}(\mathsf{pk}, m)$. This is the encryption algorithm. It takes in a message $m$ and a public key $\mathsf{pk}$ and outputs a ciphertext $\mathsf{ct}$.

- $m' \leftarrow \mathsf{Decrypt}(\mathsf{sk}, \mathsf{ct}')$. This is the decryption algorithm. It takes in a ciphertext $\mathsf{ct}'$ and a secret key $\mathsf{sk}$ and outputs a message $m'$.

- $\mathsf{ct}_f \leftarrow \mathsf{Eval}(\mathsf{evk}, \mathsf{ct}, f)$. This is the homomorphic evaluation algorithm. It takes in as input an evaluation key $\mathsf{evk}$, a ciphertext $\mathsf{ct}$, and a function $f$. Let $m$ be the message encrypted by $\mathsf{ct}$ (i.e. $m \leftarrow \mathsf{Decrypt}(\mathsf{sk}, \mathsf{ct})$). The output of Eval is the ciphertext $\mathsf{ct}_f$ that encrypts $f(m)$.

FHE must satisfy the same security level as a regular encryption scheme, which dictates that a party without access to the secret key cannot distinguish between encryptions of any two messages, even if the messages are adversarially chosen.

## 3. Approximate Activation Functions

Since the CKKS scheme is designed to handle arithmetic operations, polynomial evaluation is easily supported. In contrast, functions like $\exp(\cdot)$ or $\tanh(\cdot)$ cannot be supported in a straightforward way. Following prior work on integer-only evaluation of deep learning models (Dong et al., 2023; Zhang et al., 2024), we approximate all functions required in the LLM evaluation with low-degree polynomials. Keeping the degree low is important as this minimizes the levels consumed in the polynomial evaluation, resulting in fewer bootstrapping calls. However, if the degree is too low then the approximation may not provide good accuracy. Below we discuss the approximations of different functions we use all throughout our work. These approximations are typically parameterized by different values that determine the degree and hence the respective accuracy. We discuss in Section 4 how we instantiate these parameters concretely.

Below we point out the *depth* of the resulting computation, which is what dictates the bottleneck when instantiated with FHE. Note that a degree-$D$ polynomial can be evaluated with depth $\log_2(D)$.

## 3.1. Approximation of Comparison

We approximate the output of the sign function

$$\mathsf{sign}(x) = \begin{cases} -1 & x < 0 \\ 0 & x = 0 \\ 1 & x > 0 \end{cases}.$$

Arbitrary comparisons between $x$ and $y$ can be constructed by computing $\mathsf{sign}(x - y)$.

We use the techniques from (Cheon et al., 2020). There, the approximation is given by $h(x) = f_n^{(d_f)} \circ g_n^{(d_g)}(x)$, where $f_n(x)$ and $g_m(x)$ are carefully chosen polynomials of degree $2n + 1$ and $2m + 1$ respectively. Note that the composition requires depth $d_f \log(2n + 1) + d_g \log(2m + 1)$. We will make use of the $f$ and $g$ polynomials with degree 9 (so $n = m = 4$), and we will typically set $d_f = d_g = 2$.

## 3.2. Approximation of GeLU

We use the GeLU function (Hendrycks & Gimpel, 2016) defined as

$$\mathsf{GeLU}(x) = 0.5x \left( 1 + \tanh \left( \sqrt{2/\pi} \left( x + 0.044715x^3 \right) \right) \right).$$

As in (Zhang et al., 2024), we make use of the GeLU approximation from (Dong et al., 2023), which consists of the following:

$$\mathsf{GeLU}(x) = \begin{cases} 0, & x < -4 \\ F_0(x), & -4 \leq x < -1.95 \\ F_1(x), & -1.95 \leq x \leq 3 \\ x, & x > 3 \end{cases} \quad (1)$$

where we use the $\mathsf{sign}(x)$ approximation from above to perform the comparison. The polynomial $F_0$ has degree 3 and the $F_1$ polynomial has degree 6.

## 3.3. Approximation of Layer Normalization

Recall that the LayerNorm operation, for $\boldsymbol{x} \in \mathbb{R}^d$, is defined as

$$\mathsf{LayerNorm}(\boldsymbol{y}) := \gamma \cdot \frac{\boldsymbol{x} - \mu}{\sqrt{\sigma^2 + \epsilon}} + \beta.$$

Here $\gamma, \beta, \epsilon \in \mathbb{R}$ are constants, $\mu = \frac{1}{d} \cdot \sum_{j=1}^{d} \boldsymbol{x}[j]$ and $\sigma^2 = \frac{1}{d} \cdot \sum_{j=1}^{d} (\boldsymbol{x}[j] - \mu)^2$. The value $\epsilon$ is a fixed small constant to avoid division by zero. where $\mu = \frac{1}{n} \sum_{i=0}^{n-1} a_i$

and $\sigma = \sqrt{\frac{1}{n}\sum_{i=0}^{n-1}(a_i - \mu)^2 + \epsilon}$, where $\gamma$ and $\beta$ are learned parameters and $\epsilon$ is a small constant. The core non-polynomial operation is given by $z \mapsto 1/\sqrt{z}$, for which we can use the inverse square root uses the techniques from (Qu & Xu, 2023). We discuss these velow.

**Division by Square Root.**   The authors make use of Newton's iterative method. Once a starting approximation $y_0$ of $1/\sqrt{z}$ is chosen, iterate the following for $i = 1, \ldots, n$:

$$y_i = \frac{y_{i-1}(3 - zy_{i-1}^2)}{2},$$

with the final approximation being $y_n$. This has depth $3n$.

For choosing the initial point $y_0$, the authors first run a less accurate yet more efficient method. For this they propose two options: Taylor expansion, which is suitable for $x > 1$, and using Remez rational approximation, which is better for values that are close to 0. The work of (Zimerman et al., 2023) has found empirically that the variance (which is essentially the input to the square root) is large, so we use the Taylor expansion for the initial value.

For an approximation in the interval $[a, b]$, we choose an odd order Taylor expansion around $z_0 = (a + b)/2 + 1$ as the approximate initial value of $1/\sqrt{z}$. As suggested in (Qu & Xu, 2023), we take degree 3 (which requires depth 2), so concretely this Taylor approximation looks like:

$$z \mapsto \frac{1}{\sqrt{z_0}} - \frac{z - z_0}{2\sqrt{z_0^3}} + \frac{3(z - z_0)^2}{8\sqrt{z_0^5}} - \frac{5(z - z_0)^3}{16\sqrt{z_0^7}}$$

### 3.4. Approximation of SoftMax

For $\boldsymbol{x} \in \mathbb{R}^d$, SoftMax is defined as

$$\boldsymbol{y} = \frac{\exp(\boldsymbol{x}[i] - x_{\mathsf{max}})}{\sum_{j=0}^{m-1}\exp(\boldsymbol{x}[j] - x_{\mathsf{max}})},$$

where $x_{\mathsf{max}} = \max(\boldsymbol{x})$. The division by $e^{x_{\mathsf{max}}}$ is done in order to avoid large numerators and denominators.

**Exponentiation.**   The approximation of exp is done via Taylor series, as in (Lu et al., 2023):

$$\exp(x) \approx (1 + \frac{x}{2^r})^{2^r}, \quad x \leq 0,$$

where $r$, which corresponds to the resulting depth, is a parameter of choice.

**Max.**   Although we could compute max using an arithmetic-friendly circuit as in (Cheon et al., 2020), we proceed instead as in (Cho et al., 2024) by empirically approximating this max value—per layer—using different datasets. While the authors (Cho et al., 2024) claimed

that this optimization does not affect accuracy without any experimental evidence, in our work we verify this is the case indeed for GPT-2 small (see Table 1). This allows us to avoid the circuit for computing max while maintaining comparable accuracy to the cleartext model.

**Division.**   Division uses Goldschmidt algorithm, which works as follows. To divide $A/B$, start with an approximation $F_0$ of $1/B$, and set $N_0 = A$ and $D_0 = B$. Then iterate $F_i \leftarrow 2 - D_{i-1}$, $N_i \leftarrow N_{i-1} \cdot F_i$ and $D_i \leftarrow D_{i-1} \cdot F_i$, for $i = 1, \ldots, d$. The output of the division is $N_d \approx A/B$. The depth of this approximation is $d$, since each iteration consumes one level.

In (Even et al., 2005), it is shown that, if $0 < F_0 < 2/B$, then the algorithm converges. We set $F_0 = 10$ as the initial estimate, which works well in our experiments.

## 4. Experimental Results

In this section, we present the full LLM runtimes under FHE. These evaluations are run entirely on the server, and at no point can the server view the underlying query or any intermediate value. Furthermore, the output of the LLM forward pass can be fed directly back into the model to compute the next token without any interaction with the client. This powerful technique allows an arbitrary number of forward passes to be executed on the client's encrypted query. This method extends to other operations that require the forward-pass as a subroutine, such as fine-tuning on private data.

As we mentioned in the introduction, we focus on GPT-2 due to its accessibility as well as the similarity in the architecture of larger GPT models. We consider the small variant of GPT-2. Our runtimes can be extended to models with many more parameters by linearly scaling the transformer architecture.

### 4.1. Accuracy of the Approximate Model

In order to make our LLM compatible with FHE, we replace each non-linear function with the corresponding approximation described in section 2. We evaluate the three variants of GPT-2 on standard accuracy benchmarks to ensure that these approximations do not compromise the model's performance. We achieve this by forking the GPT-2 implementation in the HuggingFace `transformers` library (https://github.com/huggingface/transformers), and making the following modifications in order to reflect the changes that FHE imposes:

• The GeLU activation is replaced by the approximation from Section 3.2. We use degree 2 for the $f$ and $g$ polynomials in the comparison from Section 3.1, and we compose

them 2 times each.

- LayerNorm is approximated as in Section 3.3. We use 16 or 18 Newton iterations depending on model size as shown in table 2.

- SoftMax is approximated as in Section 3.4. For the approximation of exp we use $r = 7$, and for Goldschmidt algorithm—used for the division—we use 14/18/22 iterations based on model size as shown in table 2.

Performing these modifications is intricate as the `transformers` library is not intended to support changes such as replacing the SoftMax, for instance, which is rather uncommon in machine learning contexts. Once our modified model is loaded in HuggingFace's "format", we are able to leverage the Language Model Evaluation Harness library (https://github.com/EleutherAI/lm-evaluation-harness), which includes multiple benchmarks to evaluate LLM performance.

Our accuracy benchmarks appear in table 1, where we measure the performance of our modifications with respect to the baseline GPT2 - Small, GPT2 - Medium and GPT2 - Large models. We run eight diverse tasks: HellaSwag, ARC (Easy), PIQA, Social IQa, MNLI, SST-2, ANLI, and WiC.

HellaSwag tests an LLM's ability to perform commonsense reasoning about situations described in natural language. ARC (AI2 Reasoning Challenge) is a dataset created by the Allen Institute for Artificial Intelligence (AI2) to evaluate question answering systems' ability to perform multi-step reasoning. PIQA is the Physical Interaction Question Answering tasks to test physical commonsense reasoning. Social IQA measures social and emotional intelligence through questions about social interactions. MNLI (Multi-Genre Natural Language Inference) and ANLI (Adversarial Natural Language Inference) benchmark a models understanding of entailment, contradiction, and neutrality across genres and adversarially selected examples. SST-2 (Stanford Sentiment Treebank) evaluates binary sentiment classification of movie reviews. WiC (Word-in-Context) challenges models on contextual word sense disambiguation.

We also provide the different parameters we used in table 2. Together, these benchmarks provide a well-rounded evaluation of language understanding, covering reasoning, sentiment, social context, and word meaning. We refer the reader to the evaluation harness library for details on these tasks.

Overall, we observe that our modifications incur in little accuracy degradation with respect to the baseline model. This reflects the robustness of large language model to slight deviations, highly exploited in the quantization literature (*cf* (Zhu et al., 2023)), and is crucial for enabling privacy-preserving inference. Note that these approximations are also useful for MPC-based approaches.

## 4.2. Runtimes of LLM Inference in FHE

We now present the end-to-end runtime of a GPT-2 forward pass using our GPU-accelerated FHE. Note that, as illustrated in Fig. 1b, the complexity of a GPT forward pass is dependent on the position of the token being generated in the output, given that the dimension for the softmax in each decoder block depends on the token position. Furthermore, all tokens of the input sequence have to be processed *once* by the decoder blocks before any new token can be generated. Throughout this section, we benchmark generating a token at position 128, assuming that the previous input tokens have been processed. The cost of processing the input is amortized away as more tokens are produced, which is also consistent with prior works.

We note a few important optimizations that are incorporated into this benchmark:

*Input & Output Sizes.* We give the depth of each approximation in table 3. Recall from the high-level GPT architecture that SoftMax and GeLU are run once per block and LayerNorm is run twice per block. The GPT-2 model consists of 12 blocks, and the final ArgMax function is run at the end of the forward pass. The dimension of one token embedding is 768, and the inputs and outputs of both LayerNorm operations is $128 \times 768$. The GeLU input consists of 24 channels of the typical $128 \times 768$, resulting in a total input of $3072 \times 768$. By contrast, the SoftMax input is the result of many inner-product operations with the context embeddings, resulting in an input and output size of $128 \times 128$. With $2^{16}$ slots in each ciphertext, this gives the values in the second row of Table 3.

*Batched Evaluation.* When a function is evaluated over an input that does not use all available slots in a ciphertext, additional performance can be gained by evaluating another input to that function and using the additional unused slots. This batched evaluation maximizes the available parallelism in the CKKS scheme. For example, the LayerNorm function only requires 1.5 ciphertexts to store the input and output. If only one LayerNorm function is being evaluated, then we must perform the operation over two ciphertexts even though the second is half empty. However, if we have the option of running a second LayerNorm function over an independent input, we can evaluate both LayerNorm functions using only three ciphertexts, which doubles our throughput with only a 50% increase in latency. This is an important optimization for tasks such as training or fine-tuning, where the model is evaluated on batches of samples from the training set. We also present

| Model | Benchmark | ARC (Easy) | Hella-Swag | PIQA | Social IQA | MNLI | SST2 | ANLI | Wic |
|---|---|---|---|---|---|---|---|---|---|
| GPT-2 small | **Baseline** | 0.438 | 0.289 | 0.629 | 0.366 | 0.337 | 0.551 | 0.349 | 0.492 |
| | **This Work** | 0.429 | 0.281 | 0.626 | 0.374 | 0.332 | 0.556 | 0.348 | 0.511 |
| GPT-2 medium | **Baseline** | 0.490 | 0.333 | 0.676 | 0.391 | 0.351 | 0.613 | 0.349 | 0.5 |
| | **This Work** | 0.482 | 0.319 | 0.671 | 0.390 | 0.353 | 0.579 | 0.341 | 0.491 |
| GPT-2 large | **Baseline** | 0.531 | 0.363 | 0.703 | 0.395 | 0.359 | 0.501 | 0.333 | 0.496 |
| | **This Work** | 0.507 | 0.321 | 0.708 | 0.391 | 0.356 | 0.5 | 0.333 | 0.481 |

*Table 1.* Performance of various sizes of GPT-2 with our different approximations vs. the unaltered baseline. While we only benchmark GPT-2 small in FHE, these plaintext accuracy benchmarks demonstrate the overall scalability of the approximations. The softmax lookup tables were computed seperately for each model.

the "unbatched" single-input evaluation for comparison.

| | Newton | exp | Goldschmidt |
|---|---|---|---|
| GPT-2 small | 16 | 7 | 14 |
| GPT-2 medium | 18 | 7 | 18 |
| GPT-2 large | 18 | 7 | 22 |

*Table 2.* Approximation parameters for three model sizes. We use polynomials of degree 4—each composed twice—for the comparison approximations (see Section 3.1). We present values of Newton iterations for the inverse square root (see Section 3.3). For Softmax, we use the Goldschmidt algorithm for division, and $r = 7$ for the exp approximation (see Section 3.4).

**Benchmarks.** We present our benchmarks in Figure 2 and Figure 3. Both figures display the forward pass time of our encrypted GPT-2 small at position 128. All individual layer benchmarks include the internal bootstrapping time, which is interleaved within the function as needed. This machine has an Intel Xeon chip running at 2.4 GHz and 2 TB of RAM as well as an NVIDIA A100 80GB PCIe.

In Figure 2, we demonstrate the speedup of our GPU-accelerated FHE library when applied to the task of a GPT-2 small forward pass. This figure measures our GPU implementation against the out-of-the-box OpenFHE functions running on a CPU.

In the unbatched forward pass, the SoftMax function is one of the most expensive operations primarily due to the low utilization of the ciphertext. When switching to batched evaluation, the overhead of the SoftMax drops significantly ($4\times$) as well as the LayerNorm function discussed above. The GeLU function has full utilization of the ciphertexts, so the overhead with batching remains the same. The batching speedups translate into the benchmarks for the full model. Recall that the full forward pass consists of 12 blocks and an ArgMax. We do not batch the ArgMax evaluation since only a small portion of the ciphertext is left unused.

| | SoftMax | LayerNorm | GeLU | Argmax |
|---|---|---|---|---|
| depth | 22 | 13 | 17 | 272 |
| # of cts | 0.25 | 1.5 | 6 | 1 |

*Table 3.* Depths of our approximate activation functions in GPT-2 small and the number of ciphertexts required to hold the input for the 128th token. The approximations (described in section 3) have the same parameters as the plaintext circuits benchmarked in table 1. The number of slots in each ciphertext is $n = 2^{16}$. Non-integer ciphertexts indicate that not all slots are filled and batched evaluation is available in this layer.

We provide benchmarks at two different security levels depending on the application requirements. Setting the security parameter $\lambda = 128$ is standard for encryption schemes, although many applications allow a slightly weaker $\lambda = 80$. Concretely, setting $\lambda = 128$ gives us a bootstrapping routine that refreshes 20 ciphertext levels in roughly 550 milliseconds, while relaxing to $\lambda = 80$ allows a bootstrapping routine that refreshes 45 levels in under 1 second. This increase in the bootstrapping throughput is the main source of speedup.

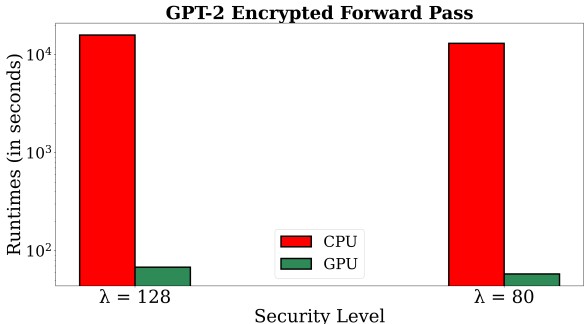

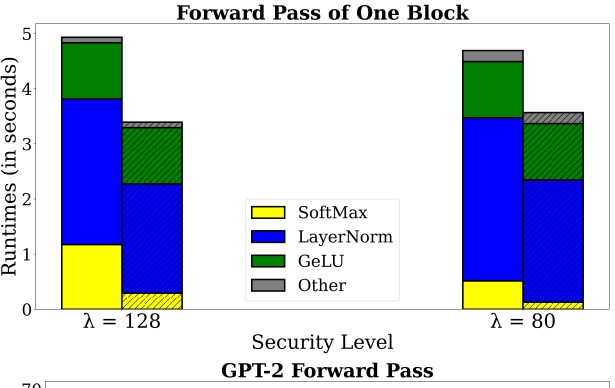

*Figure 2.* This figure presents benchmarks of our GPT-2 small forward pass running under FHE. The polynomial approximations for the activation functions as well as the high-level bootstrapping algorithm are identical in both benchmarks. The CPU bar uses the out-of-the-box OpenFHE functions, while the GPU bar uses our GPU-accelerated implementation. Both benchmarks are for a single (unbatched) evaluation. The speedup when switching to the GPU is about $200\times$.

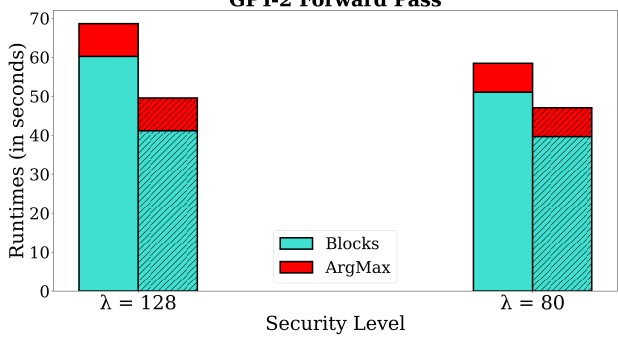

*Figure 3.* This figure presents the GPU-accelerated encrypted GPT-2 forward pass runtimes for generating a token at position 128. The hatched bars indicate the batched evaluation times, where unused ciphertext slots are filled with independent evaluations (also, see description in paragraph "batched evaluation", p.7). The savings are maximized with four independent evaluations, allowing the SoftMax to be fully utilized. The full forward pass consists of 12 blocks followed by an ArgMax.

## 4.3. Limitations

We briefly discuss the limitations of our results. Our benchmarks are based on the accuracy of the GPT-2 model with the activation functions replaced with polynomial approximations. The degree of these polynomials has a major impact on the performance of the encrypted forward pass, since a higher degree directly translates into deeper circuits that require more bootstrapping operations. While many LLM models seems to remain accurate with low precision, many other AI models such as image recognition models require higher precision during evaluation to maintain accuracy. If a model requires a higher precision than GPT-2, the polynomial approximations would need to be increased. When the required precision increases beyond roughly 16 bits, the complexity of the bootstrapping itself must be increased, since internal to the bootstrapping is an approximation of a modular reduction function. The relatively low precision required by these transformer models is crucial to our results.

## 5. Future Work

While real-time chatbots under FHE remains out of reach, these benchmarks suggest that many applications are now practical to run in a secure way. This includes tasks that do not require real-time results, such as document summarizing or drafting (e.g. "Please write a speech for our CEO."). In addition, this performance improvement can translate to tasks that require the forward pass as a subroutine, such as fine-tuning a public model on private data. This training task is computationally expensive and often requires outsourcing, which can be safely enabled by this library. More concretely, a company may wish to train a more spe-

cialized LLM for a narrow task, such as an assistant for a technical role. The additional training data for this specialized task could easily be proprietary, and the resulting model can then be decrypted by the data owner or remain encrypted on the cloud for evaluation. These applications present numerous directions for future work.

## Acknowledgments

This paper was prepared in part for information purposes by the AI Research Group, the AlgoCRYPT Center of Excellence, and Cybersecurity & Technology Controls group of JPMorgan Chase & Co and its affiliates (JP Morgan), and is not a product of the Research Department of JP Morgan. JP Morgan makes no representation and warranty whatsoever and disclaims all liability, for the completeness, accuracy or reliability of the information contained herein. This document is not intended as investment research or investment advice, or a recommendation, offer or solicitation for the purchase or sale of any security, financial instrument, financial product or service, or to be used in any way for evaluating the merits of participating in any transaction,

and shall not constitute a solicitation under any jurisdiction or to any person, if such solicitation under such jurisdiction or to such person would be unlawful.

## Impact Statement

In this work, we present a new implementation of GPU-accelerated FHE and use this implementation to evaluate an encrypted GPT-2 forward pass. This work is an important step towards practical encrypted LLM deployments, which could have extensive applications in the medical and financial industries. These industries require strict security for the sensitive data they handle, ranging from patients' medical records to corporate financial details. Enabling this data to be securely processed with state-of-the-art LLMs would significantly improve the efficiency of these industries. Furthermore, FHE is a broadly applicable technology, and accelerating FHE with commodity hardware is an active area of research. In addition to the secure LLM application discussed in this work, we view our implementation as a general advancement in the practicality of FHE.

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

## A. GPU Implementation of the CKKS FHE Scheme

In this section, we present our implementation of the CKKS FHE scheme. This implementation extends the popular and feature-rich OpenFHE library (Al Badawi et al., 2022) to use a GPU to accelerate the homomorphic operations. While we use this library to implement an LLM forward pass, this is the first open-sourced implementation of a GPU-accelerated CKKS scheme, which is of significant independent interest. The audience for this section is someone more familiar with the CKKS FHE scheme; this section can be safely skipped by those who are only interested in the LLM benchmarks. However, as a brief motivation for the focus on this function, the bootstrapping operation is at least 50% of the runtime in all layers and typically closer to 80-90% of the total time.

Our starting point for this implementation is the work of Jung et al. (Jung et al., 2021), which focuses on accelerating the bootstrapping implementation. The public portion of this code[4] is limited to the individual operations accelerated in their work, including the expensive number-theoretic transform (NTT) and RNS basis-change opera-

---

[4]https://github.com/scale-snu/ckks-gpu-c ore

tions, rather than an end-to-end bootstrapping implementation. We incorporate these kernels into the OpenFHE CKKS bootstrapping code and implement further operations to connect these core functions and avoid any data movement off of the GPU. Our code includes an end-to-end bootstrapping implementation integrated into the OpenFHE API as well as all functions required to implement the LLM layers described above. This implementation inherits the improved accuracy from the careful tracking of the CKKS scaling factor in OpenFHE.

As prior works demonstrate (de Castro et al., 2021), the bottleneck of CKKS bootstrapping quickly becomes the memory transfer if the compute accelerates faster than the local storage capacity. This is due to the size of the evaluation keys, which for bootstrapping can reach tens of gigabytes. For our benchmarks, we use a GPU with 80 GB of RAM, which allows us to cache all of the evaluation keys needed for bootstrapping and the subsequent LLM layers.

We present the benchmarks of our bootstrapping implementation in fig. 4. The CPU benchmarks were run on a machine with an Intel Xeon chip running at 2.4 GHz and 2 TB of RAM. The GPU benchmarks were run on the same machine and used an NVIDIA A100 80GB PCIeAll benchmarks were run within OpenFHE, which runs a depth 13 approximation of the CKKS modular reduction function. All bootstrapping hyperparameters were the same in all benchmarks. The level budget for the homomorphic encoding and decoding was set to 4 resulting in a total bootstrapping depth of 21. The number of decomposition digits was set to 3. The security level is at least 128 bits for the 10 and 20 output levels and 80 bits for 30 and 40 output levels. This is to accommodate the maximum modulus without growing the ring dimension.

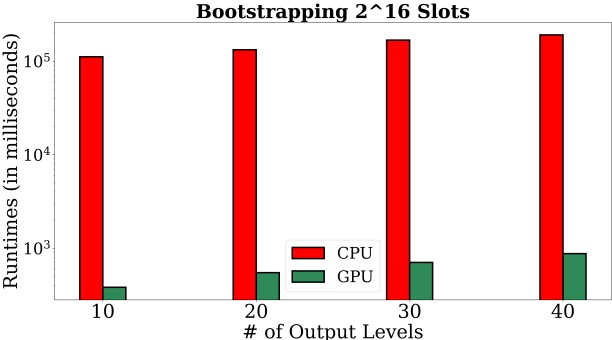

*Figure 4.* This figure presents a comparison between a CPU implementation of bootstrapping and a GPU implementation of bootstrapping for $n = 2^{16}$ slots with various output levels. Observe the log scale on the y-axis. The CPU implementation requires roughly 4-6 seconds per output level while the GPU implementation only requires 22-27 ms per output level, representing a speedup of 180-220×. These benchmarks are highly consistent with less than 5% change over 10 iterations.

