# OpenReview forum: "EncryptedLLM: Privacy-Preserving Large Language Model Inference via GPU-Accelerated Fully Homomorphic Encryption"
_ICML.cc/2025/Conference — ICML 2025 poster_

### Official Review · Reviewer_VjCM · 2025-03-11

**Overall Recommendation:** 2

**Summary:**

This privacy preservation for cloud-deployed LLMs is considered. This work proposes a GPU-accelerated Fully Homomorphic Encryption(FHE) for LLMs. Evaluations are made on a GPT-2 LLM.

**Claims And Evidence:**

See Strengths And Weaknesses below

**Essential References Not Discussed:**

See Strengths And Weaknesses below

**Experimental Designs Or Analyses:**

See Strengths And Weaknesses below

**Methods And Evaluation Criteria:**

See Strengths And Weaknesses below

**Other Comments Or Suggestions:**

1.	L804: citation error.
2.	The paper is written like a tech report for a software. I found it hard to grasp many useful insights, neither theoretical nor technical.
3.	Some more works (eg [A]) on homomorphic encryption for transformers can be discussed.


[A] HETAL: efficient privacy-preserving transfer learning with homomorphic encryption. ICML’23.

**Other Strengths And Weaknesses:**

strengthens:

1.	Promising performance (200 times faster than the CPU baseline).

2.	Source code of the implementation provided as supplemental material.

Weaknesses:

1.	The empirical evaluation is only done for GPT-2 small (124M), which is somewhat like a toy model compared to today’s mainstream open-sourced LLMs like LLAMA2 (7B/13B). it remains questionable whether the conclusion made on the former still hold for the latter.

2.	The effect of FHE approximation on the utility performance. I am afraid an evaluation on only three benchmarks can hardly capture the performance of an LLM.

3.	Lack of ablation study on the choices of implementation

**Questions For Authors:**

N/A

**Relation To Broader Scientific Literature:**

See Strengths And Weaknesses below

**Theoretical Claims:**

See Strengths And Weaknesses below

---

> ### Author Rebuttal · Authors · 2025-04-01
>
> Thank you for your review.
>
> We refer to the rebuttal of reviewer gAoN for additional benchmarks with larger models. We will include these in the next version.

---

### Official Review · Reviewer_LCvT · 2025-03-12

**Overall Recommendation:** 4

**Summary:**

This paper addresses the practical challenge of performing LLM evaluation, where clients preserve the privacy of their inputs and model owners retain privacy of the model. They propose using FHE to achieve the goal: encrypt the client inputs, perform evaluation homomorphically over encrypted data, and the results can only be decrypted by the clients who own the decryption key.
Specifically, the authors contribute a practical implementation of GPU-accelerated FHE scheme and use it to realize and evaluate an encrypted GPT-2 forward pass.
The results show that they achieve 200x faster than the CPU-based FHE implementation.

**Claims And Evidence:**

The claim that GPU-accelerated FHE-based LLM inference makes non-real-time applications practical is supported by their evaluation results that end-to-end forward pass takes several seconds using GPU acceleration.

**Essential References Not Discussed:**

N/A

**Experimental Designs Or Analyses:**

The finding that their modifications incur in little accuracy degradation with respect to the baseline model. The results are congruent with findings in the quantization literature. This alignment across different research fields reinforces the reliability of the experimental design.


page 6 col 1 line 312-313: we benchmark generating a token at position 128.
Please clarify why 128 is chosen.


page 7 col 1 line 330-333: the comparison is confusing as the larger security parameter takes less time than the smaller security parameter. Please clarify this.

**Methods And Evaluation Criteria:**

The reason why they choose FHE is well-justified, as it maintains the same client-server communication pattern as in non-private inference.
The authors' rationale for evaluating the GPT-2 model is sound. GPT-2 is open-sourced and is a representative of the transformer model.

**Other Comments Or Suggestions:**

Typos:
* page 2 col 2 line 98-100: This required computing the table of max lookup values, which we based on the extensive tests of the approximate model. (remove “we”?)
* page 7 col 2 line 372: could easily be proprietary, and The resulting model can (remove “The”?)


Comments:
* page 6 col 1 line 317-318: We note a few important optimizations that are incorporated into this benchmark
I was confused the first time I read the paragraph below cuz there’s only one optimization (batch evaluation), while “Input & Output Sizes” is a prerequisite for the optimization. And also this optimization is not FHE specific, but more like a general optimization approach.

**Other Strengths And Weaknesses:**

Strengths:
This paper is well-written. I really enjoy reading the paper. The clear explanations and structuring makes the methodology and findings accessible to readers.


Weaknesses:
* While engineering aspects are well-executed, the paper lacks substantial theoretical contributions. Most of the techniques used in this paper are quite standard, e.g., use polynomial approximations to improve the performance.
* Although the authors discuss the differences with other works in the related work section, including numerical data comparisons could significantly strengthen this work.

**Questions For Authors:**

* Please clarify how do you balance between privacy (security parameter) and utility
* Is there any engineering efforts that worth mentioned? Like how much lines have been added to the original framework?
* “LM evaluation harness library to select optimal parameters for the tradeoffs between efficiency and accuracy.”
 Could you please clarify how you decide the criteria to choose those parameters?
* page 5 col 1 line 272-274: Our runtimes can be extended to models with many more parameters by linearly scaling the transformer architecture.
Could you please explain explicitly how linear scaling would work?

**Relation To Broader Scientific Literature:**

The method can be extended to other operations that require the forward-pass as a subroutine, such as fine-tuning on private data.

**Theoretical Claims:**

This is a practical paper, no theoretical claim made.

---

> ### Author Rebuttal · Authors · 2025-04-01
>
> Thank you for your review.
>
> Q1: We set the parameters of our approximation so that accuracy is essentially unchanged. To see the evidence of how well our approximations scale, please see the rebuttal for reviewer gAoN.
>
> Q2: We added roughly 10k lines of code to the core OpenFHE library as well as an additional 5k lines for the LLM layer benchmarks & associated tests. This was a complete reimplementation of the core CKKS algorithms optimized for the high-throughput parallelism of a GPU.
>
> Q3: We chose three standard benchmarks to test our approximations, and we selected the three values in the submission as varying sufficiently from one another. In the rebuttal for gAoN, we give benchmarks on additional datasets.
>
> Q4: The CKKS homomorphic operations are highly parallel, so the runtimes of the activation function layers will mostly scale with the size of the input. The runtime of the linear layers will also grow, although proportionally this will still be a small additional overhead. Aside from minor changes due to the circuit layout, the model runtime can be computed by summing the runtimes of the individual layers.

---

### Official Review · Reviewer_BXJU · 2025-03-16

**Overall Recommendation:** 1

**Summary:**

The paper presents a novel approach to privacy-preserving inference for large language models (LLMs) using GPU-accelerated fully homomorphic encryption (FHE), specifically targeting the GPT-2 architecture. It addresses significant privacy concerns associated with LLMs, particularly when deployed on third-party cloud services, by allowing clients to encrypt their queries and perform secure computations without disclosing sensitive information. The authors introduce a GPU-based implementation of the CKKS scheme, which enhances the performance of homomorphic operations, achieving speedups of over 200 times compared to traditional CPU methods. Key contributions include efficient approximations for layer normalization and SoftMax operations, which maintain model accuracy while optimizing computational efficiency. The paper also discusses the architecture of the neural network decoder block and the use of polynomial approximations for activation functions to facilitate secure evaluations under FHE. Experimental results demonstrate the feasibility and practicality of their approach for applications such as document summarization and fine-tuning on private data. Overall, the research significantly advances the field of secure LLM inference, making it more accessible for sensitive applications in areas like healthcare and finance.

**Claims And Evidence:**

1. I think the technical contribution is very weak for this work. I think author should better clarify the challenge for using GPU to perform this large-scale task, and what is the main difference from CPU.

2, Missing important citation [1], this work also using GPU to do large-scale machine learning model inference in FHE. How is the difference between this work and [1]?

Reference:
[1] Zhang, J., Yang, X., He, L., Chen, K., Lu, W. J., Wang, Y., ... & Yang, X. (2024). Secure transformer inference made non-interactive. Cryptology ePrint Archive.

**Essential References Not Discussed:**

see above

**Experimental Designs Or Analyses:**

none

**Methods And Evaluation Criteria:**

see above

**Other Comments Or Suggestions:**

none

**Other Strengths And Weaknesses:**

none

**Questions For Authors:**

none

**Relation To Broader Scientific Literature:**

none

**Theoretical Claims:**

none

---

> ### Author Rebuttal · Authors · 2025-04-01
>
> Thank you for your review.
>
> We assure the reviewer that implementing GPU-accelerated FHE is a highly non-trivial task, requiring the synthesis of dozens of algorithms & optimizations from prior works. We additionally implemented several optimizations derived from the design of custom ASIC & FPGA implementations of CKKS. This includes the fusing of operations into CUDA larger kernels to maximize throughput. Overall, we added roughly 10k lines of code to the core OpenFHE library. We are happy to provide more details of our implementation in a comparison to the work [1] of Zhang et al. We had previously excluded [1] from our comparisons in an earlier draft of this work, since prior versions of [1] had serious errors in the reported benchmarks. The current version of [1] is the latest in a series of revisions correcting these major issues. We have not had a chance to confirm their latest results, although based on the benchmarks reported in the latest version of [1] our bootstrapping is at least 10x faster. We are confident we can outperform this work when using the same resources & parameters.

---

### Official Review · Reviewer_gAoN · 2025-03-17

**Overall Recommendation:** 2

**Summary:**

This paper presents a GPU-accelerated implementation of CKKS-based fully homomorphic encryption (FHE) for non-interactive private LLM inference. Specifically, it focuses on enabling privacy-preserving (for users' sensitive data) access to proprietary LLMs (e.g., ChatGPT) for latency-tolerant tasks such as document summarization.

To make the model compatible with HE-only inference, and also to reduce the computational burden, the authors employ off-the-shelf approximation techniques for nonlinear operations such as Softmax, GELU, and LayerNorm. The authors have shown a remarkable speedup—approximately **200×** over a standard CPU-based implementation (openFHE)—on the GPT-2 small model (12 layers, 12 heads, 768 embedding dimensions). This acceleration is primarily achieved by optimizing bootstrapping operations, which constitute the dominant source of latency in CKKS-based FHE.

**Claims And Evidence:**

Yes.

**Essential References Not Discussed:**

[1] Watson et al., Piranha: A GPU Platform for Secure Computation, USENIX Security 2022

While this paper does not accelerate FHE on GPUs but rather focuses on MPC-based acceleration for nonlinear operations, the authors should have included a discussion on whether their approach could be extended to accelerate LLM nonlinearities such as Softmax, GELU, and LayerNorm.

[2] Kim et al., Cheddar: A Swift Fully Homomorphic Encryption Library for CUDA GPUs, 2024

This paper directly focuses on GPU acceleration for CKKS-based FHE. Could the NTT acceleration presented in [2] be integrated with the authors' bootstrapping acceleration for further improvement? If so, discussing the feasibility, potential challenges, and expected benefits of such a combination would strengthen the paper and provide valuable insights for future work.

[3]  Jha et al., AERO: Softmax-Only LLMs for Efficient Private Inference, 2024

This paper removes LayerNorm and GELU activations to enable faster private inference in hybrid protocol settings (HE + MPC). Given this design choice, would the authors' GPU-accelerated approach be even more beneficial for LLMs with fewer nonlinear components? A discussion on how the acceleration scales with different levels of nonlinearity would strengthen the paper and provide insights into the broader applicability of their method.

**Experimental Designs Or Analyses:**

Yes. The experimental evaluation (presented in Table 1) is quite limited, and it could have been more comprehensive. There are two key limitations: 1) the authors have used only the GPT-2 small model, and do not show the implication of performance degradation incurred from the approximation of nonlinear operations for deeper and wider models. For example, what happens (to the efficiency gain and performance degradation) when we increase the number of layers (from 12 to 18 or 24) and/or context size (from 128 to 256); and 2) since the authors have used the pre-trained model and evaluated the performance degradation on downstream tasks, it should have been included a more diverse set of downstream tasks (from the llm-evaluation-harness library).

Ideally, the implications of approximations should also be shown for training from scratch. How much does it increase the perplexity?

**Methods And Evaluation Criteria:**

Yes.

**Other Comments Or Suggestions:**

$\bullet$ The threat model presented in Section 1.3 could be more clearly articulated. It is not explicitly stated whether the setting assumes a semi-honest or malicious client/server, which is crucial for understanding the security guarantees of the proposed approach.

$\bullet$ There is an excessive emphasis on the basics of LLM architecture and cryptographic protocols in the main text, which could have been more concise. Meanwhile, some key experimental results have been relegated to the appendix. In particular, the results in **Appendix E** are crucial to the paper’s core contributions and should be included in the main text for better visibility and impact.


**Line #385** Substituting BatchNorm with LayerNorm in LLM does not work. See [3]

[3] Wang et al., Understanding the Failure of Batch Normalization for Transformers in NLP, NeurIPS 2022.

**Other Strengths And Weaknesses:**

### Strength

$\bullet$ The authors have provided the code implementation and promised to open-source it, which could be beneficial for the researcher working in this field.

$\bullet$ Writing is coherent and the paper is easy to follow.

**Questions For Authors:**

$\bullet$ Did you include the final LayerNorm layer (in the LM-head) for end-to-end latency? A GPT-2 model with the 12 layers has 2*12 +1 LayerNorm layers.

$\bullet$ Does the GPU acceleration improve the NTT kernel?

**Relation To Broader Scientific Literature:**

It could have been much better when it comes to the connection with the prior related findings/results. For instance, the paper does not mention the some of the prior work on GPU acceleration (HE and MPC), and also the methods for improving the bootstrapping performance.

**Theoretical Claims:**

There are no theoretical claims made in the paper.

---

> ### Author Rebuttal · Authors · 2025-04-01
>
> Thank you for your review.
>
> The threat model here is semi-honest.
>
> We wrote this paper with a broad audience in mind, including cryptographers who may not be familiar with the LLM circuit. The paper describes the details of the LLM circuit as it is necessary to completely implement the LLM as a homomorphic encryption circuit, so in a sense this is a description of the homomorphic computation that is benchmarked in our work.
>
> We agree that the results in Appendix E are very important, and our implementation of GPU-accelerated homomorphic encryption is a core contribution of this work. We placed these results in the appendix since the main audience for this work is machine learning researchers who may not be familiar with the FHE bootstrapping operation. We are happy to move them to the main body.
>
> Q1: Yes, the final Layernorm time is included in the final argmax time. We will clarify this in the next version.
> Q2: Yes, all FHE operations are performed on CUDA device vectors without transferring back to the CPU. This includes all low-level polynomial operations like the NTT as well as higher-order FHE algorithms like key switching, automorphisms, and residue number system decomposition.
>
>
> Larger models:
>
> Since the submission, we have run additional experiments on larger models with nearly identical approximation parameters.
>
> All accuracy benchmarks are reported with the standard, unmodified benchmark first followed by the model run with polynomial approximations.
>
> GPT-2 Small (additional benchmarks)
>
> - Social IQA: 0.366, 0.374
> - MNLI: 0.337, 0.331
> - SST2: 0.550, 0.556
> - OpenBook QA: 0.164, 0.186
> - ANLI-R1: 0.341, 0.341
> - ANLI-R2: 0.339, 0.329
> - ANLI-R3: 0.349, 0.348
> - Wic: 0.492, 0.511
>
>
> GPT-2 Medium
>
> - Arc Easy: 0.491, 0.489
> - PIQA: 0.676, 0.675
> - Social IQA: 0.391, 0.393
> - MNLI: 0.352, 0.354
> - SST2: 0.614, 0.638
>
>
> GPT-2 Large
>
> - Arc Easy: 0.532, 0.533
> - PIQA: 0.703, 0.707
> - Social IQA: 0.396, 0.393
> - MNLI: 0.359, 0.357
> - SST2: 0.5, 0.5

---

### Decision · Program_Chairs · 2025-05-01

**Decision:**

Accept (poster)

**Comment:**

This paper proposes a method to performing FHE with GPUs that achieves significant speedup compared to existing CPU solutions. This required both significant engineering and systems design, and some novel applications of cryptographic techniques to achieve the superior performance. However, some reviewers were concerned with the lack of thorough experiments which was partially addressed in the rebuttal. Further, though this is a substantial improvement, it is likely still not the case that FHE can be deployed (even with GPU) for large frontier models.